# Evidence of cooperative effect on the enhanced superconducting transition temperature at the FeSe/SrTiO$_3$ interface

Q. Song[1,2,3], T.L. Yu[1,2,3], X. Lou[1,2,3], B.P. Xie[2], H.C. Xu[1,2,3], C.H.P. Wen[1,2,3], Q. Yao[1,2,3], S.Y. Zhang[4,5], X.T. Zhu[4,5], J.D. Guo [4,5,6], R. Peng[1,2,3] & D.L. Feng[1,2,3]

At the interface between monolayer FeSe films and SrTiO$_3$ substrates the superconducting transition temperature ($T_c$) is unexpectedly high, triggering a surge of excitement. The mechanism for the $T_c$ enhancement has been the central question, as it may present a new strategy for seeking out higher $T_c$ materials. To reveal this enigmatic mechanism, by combining advances in high quality interface growth, $^{16}$O $\leftrightarrow$ $^{18}$O isotope substitution, and extensive data from angle resolved photoemission spectroscopy, we provide striking evidence that the high $T_c$ in FeSe/SrTiO$_3$ is the cooperative effect of the intrinsic pairing mechanism in the FeSe and interactions between the FeSe electrons and SrTiO$_3$ phonons. Furthermore, our results point to the promising prospect that similar cooperation between different Cooper pairing channels may be a general framework to understand and design high-temperature superconductors.

[1] State Key Laboratory of Surface Physics and Department of Physics, Fudan University, 200433 Shanghai, China. [2] Laboratory of Advanced Materials, Fudan University, 200438 Shanghai, China. [3] Collaborative Innovation Center of Advanced Microstructures, 210093 Nanjing, China. [4] Beijing National Laboratory for Condensed Matter Physics and Institute of Physics, Chinese Academy of Sciences, 100190 Beijing, China. [5] School of Physical Sciences, University of Chinese Academy of Sciences, Beijing 100049, China. [6] Collaborative Innovation Center of Quantum Matter, Beijing 100871, China. These authors contributed equally: Q. Song, T. L. Yu Correspondence and requests for materials should be addressed to R.P. (email: pengrui@fudan.edu.cn) or to D.L.F. (email: dlfeng@fudan.edu.cn)

The 2012 discovery of an anomalously large super-conducting gap at the interface between a one-unit-cell-thick (henceforth referred to as monolayer or 1 ML) FeSe film and the SrTiO₃ (STO) substrate has set a record for interface enhanced superconductivity[1]. It is a striking realization of the hope that at the interface between two different materials one might observe the confluence of the most desirable properties from both sides.

Developments subsequent to ref. [1] have shown that by judiciously engineering the substrate the superconducting gap opening temperature can reach as high as 75 K (refs. [2,3]). Moreover, concerns that the observed energy gap might not be a super-conducting one were put to rest by converging data from angle-resolved photoemission spectroscopy[4–6] (ARPES), mutual induc-tance[7] and muon spin relaxation measurements[8]. In addition, an in situ four probe transport measurement has seen a resistivity downturn at a temperature as high as 109 K (ref. [9]). The $T_c$ of the FeSe/STO interface is consistently 20–50% higher than the highest obtainable $T_c$ in systems with nearly identical Fermi surfaces[10–14]. Such a significant enhancement of $T_c$ at the interface between the monolayer FeSe and the highly polarizable substrate has stimu-lated a surge of interest in the origin of the $T_c$ enhancement.

Several proposals have been put forward to address the role of the interface. For example, interfacial tensile strain has been proposed to enhance the antiferromagnetic exchange interaction in FeSe, thus enhancing the superconductivity[15]. However, this proposal is excluded based on the negligible change of $T_c$ in films with varied strain[2,3,16,17]. As another example, it is postulated that the enhanced $T_c$ could be the result of the huge low-temperature polarizability of STO, which results in better screening of the Coulomb interaction[18]. This proposal is also disfavored by the experimental fact that similarly high $T_c$s have been achieved for FeSe/Nb: BaTiO₃ and FeSe/TiO₂ where the substrate polarizability is very different from that of STO (refs. [3,19,20]). On the other hand, echoes of the FeSe band structure (i.e. replica or side bands) have been observed, and their separations from the main band are close to certain optical phonon energies in STO, therefore, interfacial electron–phonon interactions (EPI) were postulated to help induce the high $T_c$ (refs. [3,6,19,21]). However, the lack of direct evidence has touched off turbulent debates on whether interfacial EPI exists across the interface and how it relates to the high $T_c$ in FeSe/STO. These range from whether the replica band is simply a renormalized $d_{xy}$ band[22] or due to phonon shake-off effects, to whether the shake-off is related to initial-state effects or to the ejected photoelectrons[23]. Moreover, supposing the existence of interfacial EPI, there remains substantial debate as to whether and how interfacial EPI relates to the superconductivity[6,18,19,23–29].

Identifying the mechanism requires effective manipulation of the interfacial phonons, quantitative characterization of both the interfacial EPI and superconductivity, and effective control of film quality, which are all challenging and have not been achieved so far.

In this work, we combined advances in interface growth of superior quality FeSe/STO, $^{16}$O ↔ $^{18}$O isotope substitution, and extensive angle-resolved photoemission spectroscopy studies of many thin films. Quantitative analysis on the superconducting gaps has been carefully performed on samples with well-controlled quality and doping (see Methods). Our results pin down the existence of interfacial EPI, and demonstrate a striking correlation between the superconducting pairing strength and the interfacial EPI strength, which differs drastically from the BCS picture and provides a stringent criterion for evaluating the existing theories.

## Results

### Phonon origin of the side bands. 
To determine whether the interfacial EPI exists, we first study how the electronic structure of single-layer FeSe evolves as the interfacial STO phonons are varied through $^{16}$O ↔ $^{18}$O isotope substitution. As sketched in Fig. 1a, the comparative set of isotope substituted samples includes a 1 ML FeSe film on 60 unit cells of SrTi$^{16}$O₃ (ST$^{16}$O) film (this sample is referred to as #isotope_16) and a 1 ML FeSe film on 60 unit cells of SrTi$^{18}$O₃ (ST$^{18}$O) film (#isotope_18), both on STO substrates (see Methods, Supplementary Table 1 and Supplementary Figure 1 for growth and characterization details). The features for two Fuch–Kliewer (FK) phonons (referred to as FK1 and FK2) of the STO films (Fig. 1a) can be detected by EELS[21], which clearly indicates that the phonons FK1 and FK2 in #isotope_18 are softer than their counterparts in #isotope_16 (Fig. 1b). Note that the STO here is conducting due to vacuum annealing and Nb doping, if there were any ferroelectricity in $^{18}$O substituted STO as observed in bulk and insulating STO (ref. [30]), the electric field would be screened by the itinerant electrons and would not be likely to affect the FeSe/STO interface. In situ ARPES studies show that these two types of FeSe exhibit essentially the same band structure, with a pronounced electron band noted as $\gamma$, and two replica bands, $\gamma'$ and $\gamma^*$, which duplicate the dispersion of the main band $\gamma$ with negli-gible momentum shift (Fig. 1c). The energy separation between $\gamma$ and $\gamma'$ (noted as $E_S$) is close to the energy of the FK1 phonon (noted as $\Omega_1$), and the energy separation between $\gamma$ and $\gamma^*$ (noted as $E_S^*$) is close to the energy of the FK2 phonon (noted as $\Omega_2$) (Fig. 1e). Intriguingly, both $E_S$ and $E_S^*$ change with the $^{16}$O ↔ $^{18}$O isotope substitution in STO (Figs. 1d, e). To test the statistical significance, data collected from 18 samples show that the energy separations between $\gamma$ and $\gamma'$ are 100 ± 2 meV and 95 ± 3 meV for 1 ML FeSe/ST$^{16}$O and 1 ML FeSe/ST$^{18}$O, respectively (Supplementary Fig-ure 2). These results suggest that $E_S$ and $E_S^*$ are approximately proportional to the inverse square root of the oxygen masses (Fig. 1f). The isotope dependences of $E_S$ ($E_S^*$) and $\Omega_1$ ($\Omega_2$) con-stitute compelling evidence that the replica bands are due to shake-off excitations with the STO phonons, and exclude the scenario that the replica band is merely a renormalized $3d_{xy}$ band[22].

Two kinds of phonon shake-off processes that may give rise to the replica bands have been hotly debated recently—the initial-state effect due to interfacial EPI effects on the band structure[6,24–28] and the final state shake-off effect, that is, the energy loss of the emitted photoelectrons through the excitation of STO phonons[23]. For sample #isotope_16 (#isotope_18), the phonon energy $\Omega_1$ measured by EELS is ~94.2 meV (91.0 meV), while the energy separation $E_S$ is ~100 meV (95 meV) by ARPES (Fig. 1e). The difference between $\Omega_1$ and $E_S$ demonstrates the presence of band renormalization due to EPI[6,28], whereas the final state shake-off effect is not expected to show such a difference. Moreover, ARPES data taken with photons of different energies show that the replica band intensity is independent of the photoelectron momentum perpendicular to the sample surface, which again contradicts what the final state shake-off effect would predict (see Supplementary Note 1 and Supplemen-tary Figure 3). Therefore, our data unambiguously prove that the interfacial EPI is the cause of the replica band.

### Variation of the EPI strength. 
Theories suggest that the intensity of the replica band relative to the main band is proportional to the dimensionless electron–phonon coupling constant $\lambda$ (refs. [6,24–28]). Therefore, the fact that the intensity of $\gamma'$ is sig-nificantly higher than that of $\gamma^*$ suggests a much stronger inter-facial EPI strength between FeSe electrons and the FK1 phonon than with the FK2 phonon, which can be understood considering that a larger electric dipole field is created by FK1 due to out-of-phase vibration of the Ti atom together with all six oxygen atoms (Fig. 1a). The stronger interfacial EPI strength of FeSe electrons with the FK1 phonon would induce a larger renormalization of

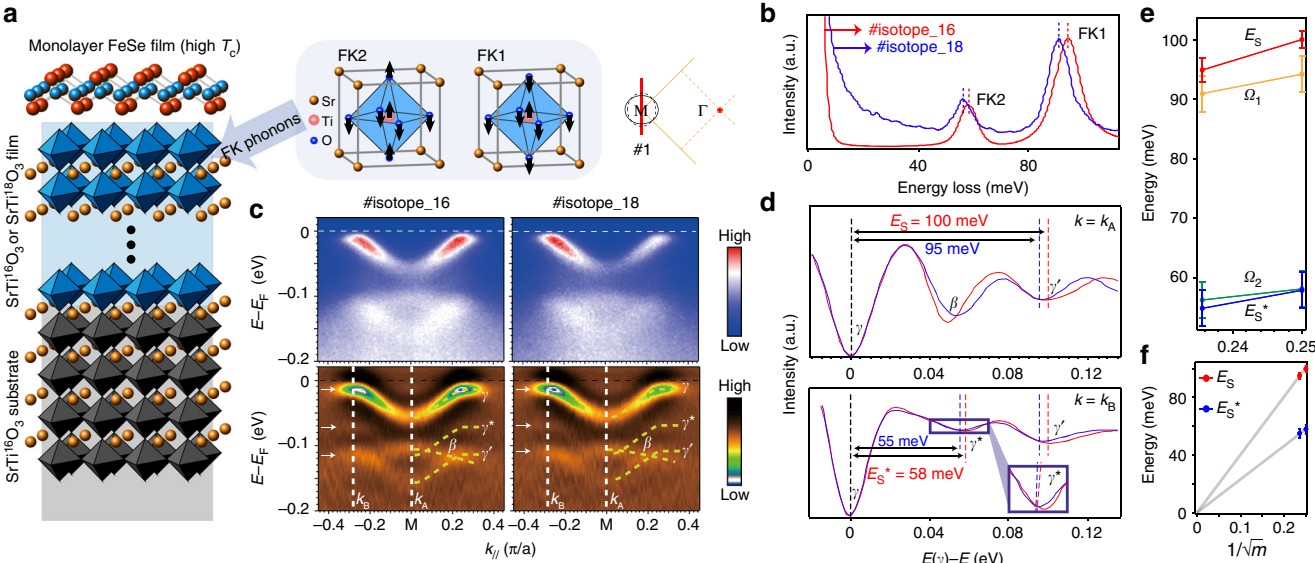

**Fig. 1** Isotope dependence of the phonon features. **a** Sketch of the single-layer FeSe film grown on 60 unit cells of either $SrTi^{16}O_3$ ($ST^{16}O$) or $SrTi^{18}O_3$ ($ST^{18}O$) film, each grown on an $ST^{16}O$ substrate. Ionic displacement patterns of the two Fuch–Kliewer (FK) phonons are illustrated. **b** Electron energy loss spectra measured by high-resolution EELS for a FeSe/$ST^{16}O$ (this sample is referred to as #isotope_16) and a FeSe/$ST^{18}O$ (#isotope_18). $\Omega_1$ and $\Omega_2$ are the phonon energies of the two FK phonons FK1 and FK2, respectively. **c** Photoemission intensity across M along cut #1 shown in the upper-middle inset and the corresponding second derivative with respect to energy to highlight the dispersions in #isotope_16 and #isotope_18. **d** Second derivative of the energy distribution curves (EDCs) with respect to the binding energy for #isotope_16 and #isotope_18 of the $\gamma$ band at the momenta $k_A$ and $k_B$ as indicated in panel **c**. $E_S$ and $E_S^*$ are the energy separations between $\gamma'$ and $\gamma$, and between $\gamma^*$ and $\gamma$, respectively, as obtained from in situ ARPES. **e** $E_S$, $E_S^*$, $\Omega_1$, and $\Omega_2$ as a function of the inverse square root of oxygen masses. **f** $E_S$ and $E_S^*$ as a function of the inverse square root of oxygen masses and the corresponding linear fits through the origin, demonstrating their proportional relationship. The error bar in energy is from the standard deviation of the data in 18 samples (Supplementary Figure 2)

the $\gamma'$ band; consistent with this, the replica band separation energy is noticeably larger than the corresponding phonon energy for $\gamma'$ while not for $\gamma^*$ (Fig. 1e). Next we focus on $\gamma'$ which represents the stronger interfacial EPI.

Extensive data were collected on high-quality samples with well-controlled electron doping and consistent single-particle scattering rate (see Methods and Supplementary Figure 9). Figure 2 shows the data measured at 6 K on six representative 1 ML FeSe/$ST^{16}O$ films, #1-#6, prepared with similar growth/annealing conditions (Supplementary Table 1) together with that of a bulk-like FeSe film (50 ML thick) under K dosing, all with carrier concentrations between 0.11 and 0.12 $e^-$ per Fe, evident from the almost identical size of the Femi surfaces (Fig. 2a). This was found to be the typical doping for FeSe/STO with 60~65 K $T_c$ (refs. [4,5]), and the optimal doping of the K-dosed FeSe thick film with 46 K $T_c$ (ref. [13]). As shown in Fig. 2b, the replica bands exist in monolayer FeSe films, but are absent in the 50 ML FeSe film whose electron-doped top layer is not affected by the interfacial EPI with STO phonons. To compare the intensities of $\gamma'$ and $\gamma$ bands, Fig. 2c displays the integrated EDCs near M, where $\gamma'$ becomes more and more pronounced from sample #1 to #6, and the $\gamma'$ spectral weight increases monotonically relative to that of $\gamma$ (Fig. 2c). We confirm that such fading of replica band intensity from sample #6 to #1 is neither due to single-particle impurity scattering (Supplementary Figure 5) nor due to the smearing out of the spectral weight (Supplementary Figure 6). Since the intensity ratio between $\gamma'$ and $\gamma$ reflects the interfacial EPI constant $\lambda$ between the FeSe electrons and FK1 phonon[6,24–28], the variation of the replica band intensity ratio suggests the variation of interfacial EPI among sample #1–#6. EPI variations in samples with the same carrier density and chemical composition would be unusual for bulk materials, but our results show that such an EPI variation can actually happen at the interface. The origin may be

related to slight variations in the STO surface leading to different interfacial bonding conditions, such as differences in bond disorder between FeSe and STO (ref. [31]). Our results call for future studies combining ARPES with interfacial atomic-scale structure characterization to resolve the detailed atomic registration and bonding conditions.

**Superconducting gap variation.** The photoemission spectra across M for these samples all show Bogoliubov quasiparticle dispersions with the opening of superconducting gaps in the two electron bands (Supplementary Figure 7). Superconducting gap anisotropy along the elliptical Fermi surfaces has been reported in FeSe/STO/KTaO₃ and FeSe/STO films[2,32] as sketched in Fig. 3a, and consistent with this, the observed superconducting gap at $k_1$ ($\Delta_1$) is smaller than the superconducting gap at $k_2$ ($\Delta_2$) in each sample (inset of Fig. 3a and Supplementary Figure 7). The EDCs measured at $k_1$ (Fig. 3b) were symmetrized and fitted to superconducting spectral functions to get the superconducting gap sizes[33–35] (Fig. 3c, d). The gap uncertainty can be reduced to 0.35 meV as the gap sizes measured at equivalent $k_F$'s are averaged (Supplementary Figure 8). The averaged $\Delta_1$ is smallest in the K-dosed FeSe thick film, while it increases from 9.3 meV to 12.1 meV (~30% variation) from K-dosed FeSe thick film through monolayer FeSe #1–#6, with increasing replica band intensity ratio. A similar trend has been observed for the superconducting gap at $k_2$ ($\Delta_2$), which increases from 9.45 meV to 13.3 meV (~41% variation, Fig. 3e, f), and such variations are significantly beyond the experimental uncertainty.

Sharp coherence peaks in the ARPES data indicate the homogeneous nature of the electronic states and low defect concentrations[35,36]. It should be noted that both the replica band features (Figs. 2b, c) and the superconducting coherence peaks (Fig. 3b) are more pronounced than previously reported[3–6,19],

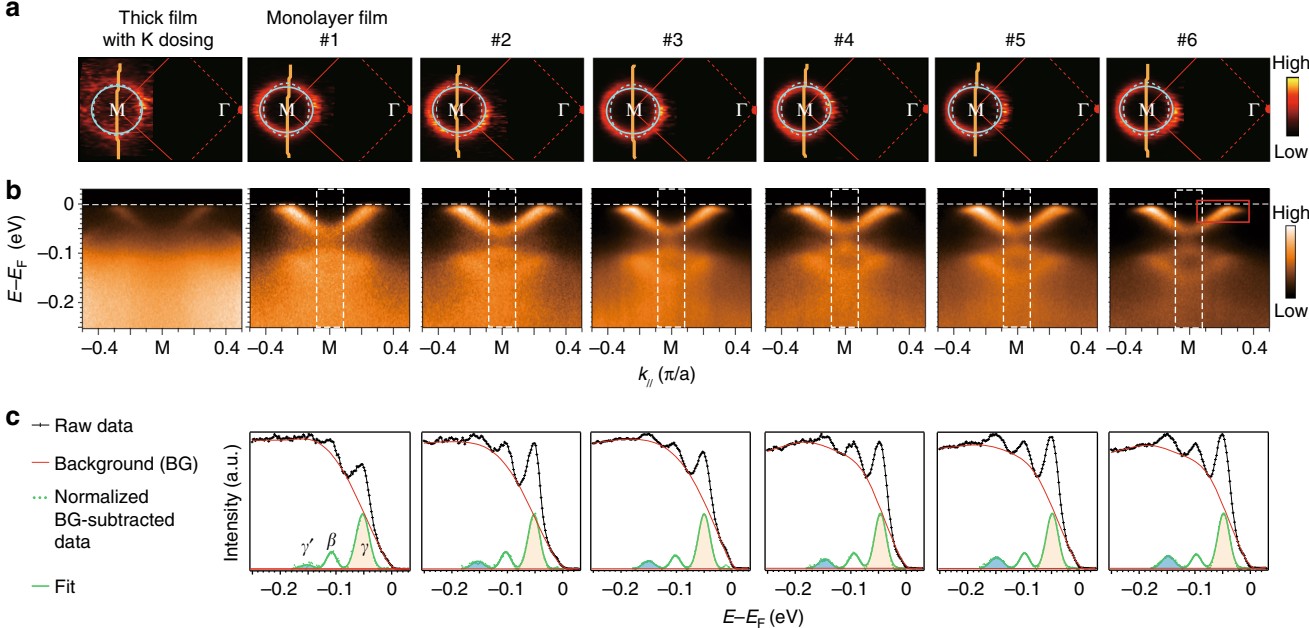

**Fig. 2** Side-band intensity in FeSe thick film and various 1 ML FeSe/ST$^{16}$O samples. **a** Photoemission intensity maps at the Fermi energy, and **b** photoemission intensity distributions across M along the cut illustrated in panel **a**, of various samples, including a K-dosed thick film and 1 ML FeSe/ST$^{16}$O samples #1–#6. **c** EDCs around M, the background used in intensity analysis, background-subtracted photoemission intensity normalized by the peak height of the $\gamma'$ band, and the corresponding fits for samples #1–#6. For better statistics, the EDCs are integrated over the momentum range indicated by the white dashed rectangle in panel **b**. The background is modeled using a cubic spline interpolation. The data (dots) are fitted to three Gaussian peaks, representing the spectral weight from the $\gamma$, $\beta$, and $\gamma'$ bands ($\gamma^*$ and $\gamma'$ have low spectral weight and are neglected here). Details of the background modeling and fitting are shown in Supplementary Figure 4. The spectral weights of bands $\gamma$ and $\gamma'$ are denoted as $I_0$ and $I_1$, respectively. The variation of the replica band intensity ratio is not related to the slight variation of the background (Supplementary Figure 5). As the intensity ratio $I_1/I_0$ decreases from sample #6 to sample #1, the replica band $\gamma'$ always has the same full-width at half-maximum as that of the main band $\gamma$ and does not smear out (Supplementary Figure 6). All data were measured at 6 K

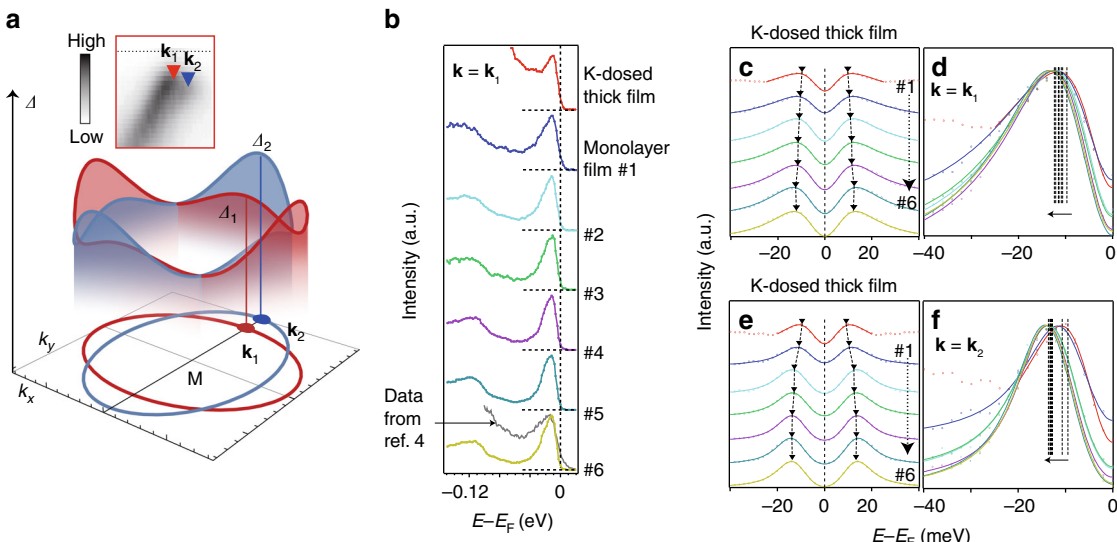

**Fig. 3** Superconducting gap variation. **a** Illustration of the electron pockets and the anisotropic superconducting gap structure. The momentum locations of $\mathbf{k}_1$ and $\mathbf{k}_2$ and the superconducting gap $\Delta_1$ at $\mathbf{k}_1$ and $\Delta_2$ at $\mathbf{k}_2$ are also illustrated. The inset is a zoomed-in spectrum of the red square part of sample #6 as shown in Fig. 2b, which indicates the two different gap sizes at the two normal state Fermi momenta $\mathbf{k}_1$ and $\mathbf{k}_2$. **b** EDCs of different samples at the Fermi momentum $\mathbf{k}_1$ compared with that of the sample with $T_c = 60 \pm 5$ K in ref. [4] (grey curve). Temperature-dependent studies show that the gap of FeSe/ST$^{16}$O with $\Delta_1 = 12.1$ meV closes at $64 \pm 4$ K (Supplementary Figure 10b, d). **c** Symmetrized EDCs at $\mathbf{k}_1$ of different samples (dots) and the fitting results to a superconducting spectral function (solid lines, see Supplementary Figure 8 and Methods). **d** Expanded view of fitted curves from panel **c** overlaid to show the gap variation more clearly. **e, f** Same as **c** and **d** but for EDCs at $\mathbf{k}_2$ and gap $\Delta_2$. All data were measured at 6 K

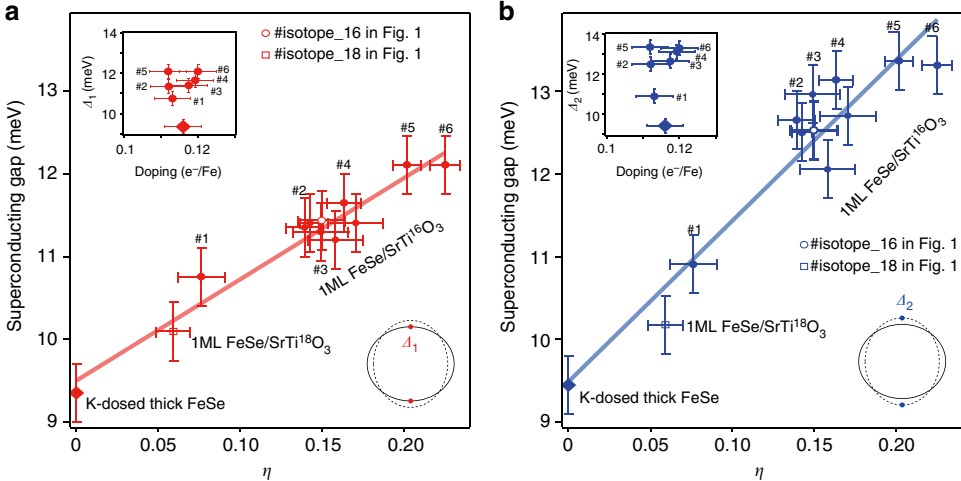

**Fig. 4** Superconducting gap as a function of electron–phonon coupling strength. **a** Superconducting gap sizes $\Delta_1$ and **b** $\Delta_2$ are plotted as a function of the intensity ratio between the side band $\gamma'$ and the main band $\gamma$, i.e., $\eta = I_1/I_0$, which is proportional to the interfacial EPI constant[6,24,25]. The red and blue bars are linear fits to $\Delta_1$ and $\Delta_2$, respectively. The insets show the superconducting gap sizes as a function of doping for representative samples. Doping variation is minimal, while the variation in gap sizes does not correlate with the doping. Note: here we use $\eta = I_1/I_0$ to represent the electron–phonon coupling constant according to theory[6,28], while using $I_1/(I_1 + I_0)$ would only slightly change the x axis and does not affect the conclusion. The error bar for $\eta$ is from the standard deviation of the fits. The error bar for the superconducting gap is described in Supplementary Figure 8

and the coherence peak intensity relative to the background is significantly larger than that in previous reports[4–6] (the gray curve in Fig. 3b from ref. [4], for instance), indicating the superior quality of our samples. Special attention has been paid when analyzing such small superconducting gap variations, and it is confirmed that the variations in superconducting gap among our samples is not due to the gap anisotropy, aging effects, charging, variations in electron doping, or variations in single-particle impurity scattering rates (see Methods). Specifically, quantitative analysis on the single-particle scattering shows that the films here have nearly identical quality. In any case, slight variations of single-particle scattering rate are not correlated to the changes in superconducting gap size in these samples (Supplementary Figure 9).

**Role of interfacial phonon in superconductivity**. To investigate the relation between superconductivity and interfacial EPI, Fig. 4 plots the superconducting gap as a function of the side-band intensity ratio $\eta = I_1/I_0$, including data points from the #isotope_16 and #isotope_18 samples in Fig. 1, six representative samples in Fig. 2, K-dosed thick FeSe film and additional high-quality samples (see Methods, Supplementary Figure 11 for a comparison between a sample with a weaker band intensity feature and #6, and Supplementary Figure 9 for spectra of all these samples). The key results of our study are summarized in Fig. 4: both the superconducting gaps $\Delta_1$ and $\Delta_2$ of the monolayer FeSe film are highly correlated with the replica band intensity ratio, which is proportional to the interfacial EPI constant $\lambda$ according to theories which consider the replica band to arise from EPI[6,24,25]. Considering that the superconducting gap size at $T \ll T_c$ is a direct characterization of pairing strength, and is proportional to $T_c$ given the essentially identical single-particle scattering rate (Supplementary Figures 9 and 10), our results directly demonstrate that superconductivity is related to EPI strength.

**Discussion**

The gap increases with $\lambda$ in a remarkably linear fashion inconsistent with the usual BCS behavior (Supplementary Figure 12), in

a form that was actually suggested to be a hallmark of the pairing-enhancement scenario where electron–phonon interactions are strongly peaked in the forward-scattering ($q = 0$) direction[6,24–28]. The extrapolation to the $\eta = 0$ limit gives an intercept around 9.5 meV for both $\Delta_1$ and $\Delta_2$ (Fig. 4), which coincides with the superconducting gap of a heavily electron-doped FeSe monolayer obtained through K dosing on the surface of a FeSe thick film[13]. That is, in the limit of zero interfacial electron–phonon coupling, FeSe/STO is closely similar to the K-dosed surface FeSe layer of a bulk-like film, whose intrinsic pairing mechanism can generate a sizeable superconducting gap already, while the interfacial EPI accounts for a linear enhancement on top of that. Our data thus directly demonstrate that the high $T_c$ in FeSe/STO is caused by the collaboration between an intrinsic mechanism and the interfacial electron–phonon interactions[6,24,25], and excludes the theories that EPI directly induces the high $T_c$ of FeSe/STO (ref. [28]). The data point of sample #isotope_18 roughly follows the relation of gap versus interfacial electron–phonon coupling strength of all the $^{16}O$ samples, indicating no observable isotope effect on the super-conducting gap (Fig. 4). This can be explained by an isotope coefficient $\alpha < 0.5$ due to multi-channel pairing[28]. Moreover, a recent theory based on a forward-scattering EPI mechanism predicts the superconductivity should be mass-independent to leading order[37]. Therefore, in these frameworks the gap variation with isotope could be too small to be resolved.

The comprehensively and quantitatively resolved parameters in our experiments enable further quantitative examination of the theoretical predictions[6,28]. For example, in ref. [28], the theory based on perfect forward-scattering EPI gives two explicit relations—$E_S/\Omega_{ph} = 1 + 2\lambda + O(\lambda^2)$ and $\eta = \lambda + O(\lambda^2)$, which however produce inconsistent $\lambda$ values of 0.032 and 0.151 from our data (based on the data from sample #isotope_16), respectively. On the other hand, it can be shown that finite momentum width of the forward scattering would reduce the band renor-malization effect significantly, thus $\lambda$ could be much larger than 0.032 here[28]. Therefore, our data provide explicit constraints for further theoretical development toward a better understanding of forward-scattering EPI.

Finally, beyond solving the mystery of the high $T_c$ in 1 ML FeSe/STO as being a collaboration between the intrinsic pairing of heavily electron-doped FeSe and the interfacial electron–phonon coupling, our results directly establish that electron–phonon interactions, particularly the forward-scattering type, can play a critical role in the high $T_c$ of a highly correlated superconductor. In a broad picture, it has been suggested theoretically that electron–phonon forward scattering can collaborate with various spin and orbital fluctuations to enhance superconductivity, be it s-wave or d-wave pairing[24]. The electron–phonon forward-scattering mechanism may be applied to a broad range of superconducting materials[38]. Since there are oxide charge reservoir layers alternating with superconducting layers in many cuprate and iron-based superconductors, it would be intriguing to search for analogous interfacial effects based on the framework established here. Overall, our data suggest a route forward for the development of interfacial-enhanced high-$T_c$ superconductors and the understanding of high-$T_c$ superconductivity in general.

## Methods

**Oxygen-isotope-substitution and sample preparation**. We prepare pure $ST^{18}O$ films on top of commercial $ST^{16}O$ substrates in three steps:

First, partially substitute $^{16}O$ in commercial $ST^{16}O$ substrates by $^{18}O$. The etched STO(001) substrates were annealed under a high vacuum of about $1.2 \times 10^{-8}$ mbar for 2 h at 750–800 °C. After the vacuum annealing, the substrates were annealed at the same temperature under a partial pressure of $1.5~2.0 \times 10^{-6}$ mbar of $^{18}O_2$ for another 2 h. The outlet of the gas injector points directly at the substrate from a distance of 6 cm, thus the actual $^{18}O_2$ pressure at the substrate should be orders of magnitude higher than measured.

Second, grow $ST^{18}O$ thin films atop the $^{18}O$ partially substituted $ST^{16}O$ substrates. Sixty unit cell $ST^{18}O$ thin films were grown layer by layer at about 700 °C under an oxygen partial pressure of about $8.0 \times 10^{-7}$ mbar in $^{18}O_2$.

Third, after the growth of $ST^{18}O$ thin films, the samples were heated to ~780 °C for 45 min in an $^{18}O_2$ partial pressure of $3.5 \times 10^{-7}$ mbar, which is lower than the growth pressure. Annealing in low pressure $^{18}O_2$ not only induces oxygen vacancies which are necessary for the epitaxy of high-quality single-layer FeSe, but also increases the $^{18}O$ concentration at the surface of $ST^{18}O$.

Secondary ion mass spectroscopy (SIMS) was performed on a 20 nm Se/60-unit-cell $ST^{18}O$/ $ST^{16}O$ sample to ensure that the STO films contain a substantial amount of $^{18}O$ (Supplementary Figure 1).

**None-oxygen-isotope-substitution sample preparation**. To obtain $ST^{16}O$/$ST^{16}O$ with identical surface quality and oxygen vacancy concentration as $ST^{18}O$/$ST^{16}O$, we anneal the substrates, grow 60 unit cells of $ST^{16}O$, then anneal these under the same conditions as $ST^{18}O_3$/$ST^{16}O$ but in an $^{16}O_2$ atmosphere. After the preparation of the $ST^{18}O$ or $ST^{16}O$ surface, the samples were transferred under ultra-high vacuum for FeSe growth. Single-layer FeSe films were grown at ~520 °C by co-evaporation of Se and Fe and then post-annealed at ~546 °C for 5.5–8.5 h. Thick FeSe films were grown at 370 °C then post-annealed at 410 °C in vacuum for 2.5 h. Surface potassium dosing is conducted with a commercial SAES alkali dispenser.

The commercial 0.5%wt Nb-doped $ST^{16}O$ substrates are from Hefei Kejin Materials Technology Co., Ltd. To avoid charging effects, the $ST^{16}O$ and $ST^{18}O$ grown on commercial substrates are doped with ~0.7%wt Nb.

**Growth of single-layer FeSe and the annealing details for sample #1–#6**. After the high-temperature annealing in a low partial pressure of oxygen, the STO substrates were transferred under ultra-high vacuum to another MBE chamber, where FeSe films were grown. Single-layer FeSe films were grown at ~520 °C by co-evaporation of Se and Fe. After growth, the films were annealed in vacuum for several hours before ARPES measurement. The growth and annealing parameters vary slightly for S16 #1–#6, and the details are listed in the Supplementary Table 1.

The FeSe growth and annealing involve much lower temperature heating ($T \sim$ 520 °C and $T <$ 548 °C respectively) in high vacuum ($P <$ 5E–9 mbar). The oxygen partial pressure is very low, and there is no possibility that the residual $^{16}O$ in the vacuum could produce any effective substitution of the $^{18}O$ in the STO during the annealing process. EELS studies were performed after all the long-time annealing before ARPES measurements and an additional annealing at 450 °C for 6 h to remove the Se capping layer in the EELS preparation chamber, and the results clearly show the different phonon energies between $^{16}O$ and $^{18}O$ samples. The isotope dependence of $\Omega$ and $E_S$ ($E_S^*$) can only be explained if the interface retains a considerable level of isotope substitution after FeSe growth and annealing.

**ARPES measurements**. The in-house ARPES measurements were performed with Fermi Instruments discharge lamps (21.22 eV He-Iα light and 10.02 eV Kr light) and a Scienta DA30 electron analyzer. The overall energy resolution is 7.5 meV, and the angular resolution is 0.3°. Samples were measured under an ultra-high vacuum of $5 \times 10^{-11}$ Torr. The sample growth, K dosing and ARPES measurements were all conducted in situ. The samples used to analyze the replica band ratio $\eta$ are all around the optimal doping level, measured exactly at the M point within ±0.5° and have similar single-particle scattering rates, avoiding differences due to doping level, gap anisotropy or sample quality.

**EELS measurements**. Single-layer FeSe/$ST^{18}O$/$ST^{16}O$ and single-layer FeSe/$ST^{16}O$/$ST^{16}O$ samples were capped by amorphous Se to protect the surface from atmosphere. The capped samples were transferred to a high-resolution EELS system, and annealed at 450 °C for 6 h to remove the Se capping layer. LEED patterns were collected to confirm the removal of the capping layer and verify the sample quality. High-resolution EELS measurements were performed at 35 K, with an incident beam energy of 110 eV and an incident angle of 60° with respect to the surface normal. The energy resolution is 3 meV.

**Determination of the superconducting gap and single-particle scattering rate**. The superconducting gap is determined by fitting the symmetrized EDCs to a superconducting spectral function with the simplified BCS self-energy $\Sigma(\mathbf{k}, \omega) = -i\Gamma_1 + \Delta^2/[\omega + \epsilon(k) + i\Gamma_0]$, in which $\Gamma_0$ is the inverse pair lifetime, which is 0 in the superconducting state, while $\Gamma_1$ can represent the single-particle scattering rate in real materials. This method of fitting the superconducting gap has been used in various cuprates and Fe-based superconductors, and gives reliable results[33–35].

It should be emphasized that photoemission data at 6 K in our high-quality samples give negligible background near $E_F$ (Supplementary Figures 9b, c), and the constant backgrounds from gap fits are all close to zero.

**Criteria of the quantitative analysis on the relation between gap size and EPI strength**. Special caution should be paid when comparing the superconducting gap sizes, especially considering that the total superconducting gap variation is quite small, within 4 meV in our experiment. In the quantitative analysis of the superconducting gap in our manuscript, we have been careful to exclude any other factors that may affect the superconducting gap size, including the gap anisotropy, aging effects, charging effects, electron doping variations, sample defect variations, etc.

To avoid any influence on the gap size from gap anisotropy, we precisely determined the momenta $\mathbf{k}_1$ and $\mathbf{k}_2$ based on the photoemission intensity map on each sample and measured $\Delta_1$ and $\Delta_2$.

To exclude any influence from aging effects, data were collected under the same conditions repeatedly every 5 min, and only the data that exhibited no aging effects were summed to get the high-statistics scan around M.

To avoid any influence from charging effects, we checked the superconducting gap size measured using the usual photon flux and using 10% of that. Any shift of the gap by 0.1 meV would cause the sample to be abandoned. All data shown in the main text are free from charging effects.

Electron doping level is another factor that could affect the superconducting properties. All the samples used for superconducting gap studies in the main text show nearly identical doping levels, and our data show that small variations in doping level cannot account for the change of gap size in our study (Insets of Fig. 4).

The sample quality of the FeSe films could affect the superconducting properties and broaden the superconducting peak. Impurity scattering can drastically reduce the lifetime of the quasiparticles, resulting in broadening of the ARPES lineshapes and diminishing of quasiparticle peak intensity. In order to exclude any influence from sample defects in the FeSe layer, all the samples used in superconducting gap studies in the main text have superconducting coherence peaks with almost identical width and similar scattering rates $\Gamma_1$ from fitting (Supplementary Figure 9). Their width and similar scattering are significantly smaller than that of the sample used in ref. [4], whose $\Gamma_1$ is about 28.3 meV, indicating the high homogeneity and low defect densities in our current samples. The larger $\Gamma_1$ from lower sample quality makes the coherence peak appear to shift to higher binding energy, which explains the overestimated gap size in previous reports. Moreover, there is no correlation between $\Gamma_1$ and the superconducting gap sizes in the samples here (Supplementary Figure 9). This further excludes the possibility that the gap variation in this work is due to defects in the FeSe layer.

Quantitative analysis of the EPI strength also requires superior sample quality and careful analysis.

The background subtraction is carefully done on each sample. The fixed points are chosen at energies and momenta without any main band or replica band features (Supplementary Figure 4). The background intensity is similar among the samples and its small variation is not correlated with either the superconducting gap or the quasiparticle scattering rate $\Gamma_1$, thus the background subtraction can give us the replica ratio reliably (Supplementary Figure 5).

Another critical factor for obtaining reliable results on the EPI strength is to have samples with pronounced main band features. Here we compare a previous

sample with $T_c \sim 60 \pm 5$ K but lower quality and weaker band intensity and the #6 sample in our paper (Supplementary Figure 11).

First, we can see that the simulation with $\eta = 0.22$ agrees with the previous data (Supplementary Figure 11a, b). Therefore, the different $T_c \sim 65$ K samples exhibit almost the same $\eta \sim 0.22$. This confirms that the relation between superconductivity and electron–phonon coupling strength is fixed for both the current high-quality samples and also previous samples.

Second, when the intensity of the main band feature is weak, as in our previous $T_c \sim 65$ K sample (Supplementary Figure 11a, b), since the replica band intensity is roughly one order of magnitude lower than that of the main band, quantitative analysis on the intensity ratio would be significantly affected by noise and cannot give reliable results. In our recent high-quality samples, the intensities of the band features show a significant enhancement compared with the background intensity, which benefits our quantitative analysis (Supplementary Figure 11c, d).

Therefore, our conclusion on the relation between gap size and EPI strength is not only solid for the data on the current good samples, but also consistent with the previous data. Comparing samples with different quality, one need to consider two factors—the sharpness of the main band will significant affect the accuracy of the quantitative analysis on the replica band intensity ratio, and large impurity scattering ($\Gamma_1$) would affect the determination of the superconducting gap size. Only after one obtains samples with superior and identical single-particle scattering rate, which show small $\Gamma_1$ and pronounced band features, can one conduct the quantitative analysis on the replica band intensity ratio and superconducting gap, and claim their relation.

## Data availability

All data were processed in Igor Pro 6.22A. Relevant data supporting the key findings of this study are available within the article and its Supplementary Information files. All raw data generated during the current study are available from the corresponding author on reasonable request. The source data underlying Figs. 1e, f, 4, and Supplementary Figures 6b, 9d-e, 10d, 12 are provided as a Source Data file.

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

## Acknowledgements

This work is supported in part by the National Key R&D Program of the MOST of China (Grants Nos. 2016YFA0300200, 2017YFA0303004) and the National Science Foundation of China (Grants Nos. 11704073, 11504342). We appreciate helpful discussions with Profs. Dragan Mihailovic, Hugo Keller, Annette Bussmann-Holder, Frederick Walker, and Victor E. Henrich on the isotope effects; and Guangming Zhang, Qianghua Wang, and Steve Johnston on EPI. We thank Dr. Darren Peets for help with the editing, and Dr. Qiuyun Chen for help with the SIMS measurements.

## Author contributions

Q.S., T.L.Y., X.L., C.H.P.W., Q.Y., B.P.X. and H.C.X. conducted film growth and ARPES measurements. R.P., Q.S., T.L.Y., and D.L.F. analyzed the ARPES data. S.Y.Z., X.T.Z. and J.D.G. conducted EELS measurements. R.P. and D.L.F wrote the paper. D.L.F. is responsible for project direction, planning, and infrastructure.

## Additional information

**Competing interests:** The authors declare no competing interests.

