## [Peer Review File · Nature Communications]

Reviewers' comments:

Reviewer #1 (Remarks to the Author):

The authors of the manuscript performed comprehensive experiments on 1 ML FeSe/STO in an effort to investigate the role of the STO phonon in the superconductivity. Especially they studied isotope effects to obtain decisive evidence for the role of the phonon. In the work, they showed that the T_c change is related to the oxygen mass but that the T_c does not follow the prediction of the BCS theory. Such observation suggests that the pairing is not directly mediated by the STO phonon but the role of the phonon is to boost the superconductivity

The issue on the enhanced superconductivity in 1 FeSe/STO is still one of the most important issues in the field of iron based superconductivity. The study reported in the manuscript is very systematic and includes very high-quality samples and data. They also provide experimental data that are needed to verify all their claims. Therefore, I would recommend the manuscript for publication but would suggest to revise the manuscript to reflect the following comments.

1. (Minor) Page 3, 1st paragraph : There should be some consistency in the notation of the oxygen isotope. It is written as ^{16}O and O^{16} .
2. Page 5, 1st paragraph: Regarding the claims of "In the supplementary information we confirm that such fading of replica band intensity from sample #6 to #1 is neither due to sample quality (Fig. S5) nor due to the smear out of the spectral weight (Fig. S6)" and "The origin may be related to slight variations in interfacial bonding conditions, such as differences in bond disorder between FeSe and STO (ref. 30)", the authors analyzed the background intensity in Fig. S5 and showed that there is no correlation between the background intensity and SC gap size, in an attempt to show that the relatively low quality ARPES data in Fig. 2 is not from simple sample quality problem. However, bond disorder may be regarded as part of sample quality. In fact, the data in Fig 2 seems to be dependent on such disorder. Therefore, the claim that the data quality variation is not from the sample quality may not be technically right. Authors may need to reword their claim as it may confuse the readers.
3. Page 8, 1st paragraph: "if there is an oxygen isotope effect on the gap as in the usual BCS formula, it is presumably less than ~ 0.2 meV.", In obtaining the value of 0.2 meV, they used $\Delta_{\text{interface}}$. $\Delta_{\text{interface}}$, however, is not clearly defined. I assume it is the gap difference between K-dosed and #6 sample but it should be defined. In addition, how do authors know if the expression used in obtaining 0.2 meV is reasonable? Any reference for it would be useful.

Reviewer #2 (Remarks to the Author):

This manuscript seeks to address the question of where electron phonon coupling at the interface between FeSe films and the chosen substrate serves to enhance the development of higher T_c superconductivity in the chalcogenide film. By using isotopic substitution the authors provide compelling evidence that that may well be the case in what appears to be a careful study. My feeling is that the manuscript can be published but I am left with some questions.

1. I was intrigued by the observation that the intensity of the phonon sidebands did not appear to have the Poisson distribution associated with Franck-Condon like processes.
2. The preparation of the samples with different isotopes involved considerable annealing, even before

the final ARPES measurement. Are the authors sure that they still had the same isotopically substituted substrate?

3. The superconducting gaps associated with O16 and O18 samples differed by maybe 2.0 meV. The energy resolution in the experiment is 7.5 meV according to the authors. It is not impossible to resolve such a small difference with that resolution but it is on the edge.

The English could be improved in places. For instance I was not sure how to interpret the statement "Since the intensity ratio between g' and g reflects the interfacial EPI constant I between the FeSe electrons and FK1 phonon⁶, 21-25, our results suggest the variation of interfacial EPI among sample #1-#6." in the first paragraph of page 5.

Further the figure captions are not always clear in their message. They need some work to make them easier to read.

Reviewer #3 (Remarks to the Author):

The authors have performed systematic measurements of the variation in electron-phonon coupling strength in several FeSe/STO samples. They show unambiguously that e-ph coupling is related to the observed 'replica' bands seen in ARPES, and that the e-ph coupling linearly enhances the superconducting gap. This provides strong evidence that besides the doping and the intrinsic bulk FeSe superconducting mechanism, electron-phonon coupling is relevant in monolayer FeSe on STO. I think the paper should be published quickly as it is extremely timely and relevant to the field, and will interest the whole superconductivity community.

I think there is only major issue that needs to be addressed more carefully. The authors have 6 samples with an oxygen-16 substrate, with "slightly different growth/annealing conditions", leading to a "variation of interfacial EPI". What is the origin of this? How can a few degree change in annealing temperature lead to a dramatic change EPI (of more than a factor 2)? Can the authors elaborate on this point?

Some minor issues:

1. In the discussion the authors mention that the oxygen isotope effect on T_c should be unobservable. I think it is worth mentioning that in the forward scattering mechanism, the isotope effect on T_c is actually to leading order absent, see for example Kulic&Dolgov in <http://iopscience.iop.org/article/10.1088/1367-2630/19/1/013020/meta>.

2. As for isotope substitution: It is exciting to see that they found a change in the replica band location upon substitution of oxygen-16 to -18 in only the top layers of the substrate. I do wonder to what extent the possible ferro-electric transition in STO, that exists upon oxygen substitution in the bulk (Rowley Nature Physics 10, 367 (2014)), is relevant for these samples. Can the authors elaborate on this relation?

Reviewers' comments:

Reviewer #1 (Remarks to the Author):

The authors of the manuscript performed comprehensive experiments on 1 ML FeSe/STO in an effort to investigate the role of the STO phonon in the superconductivity. Especially they studied isotope effects to obtain decisive evidence for the role of the phonon. In the work, they showed that the T_c change is related to the oxygen mass but that the T_c does not follow the prediction of the BCS theory. Such observation suggests that the pairing is not directly mediated by the STO phonon but the role of the phonon is to boost the superconductivity

The issue on the enhanced superconductivity in 1 FeSe/STO is still one of the most important issues in the field of iron based superconductivity. The study reported in the manuscript is very systematic and includes very high-quality samples and data. They also provide experimental data that are needed to verify all their claims. Therefore, I would recommend the manuscript for publication but would suggest to revise the manuscript to reflect the following comments.

Response: We appreciate the reviewer's positive comments and strong recommendations. We also thank the reviewer for his/her constructive comments, which helped us to improve our manuscript. Below we list our detailed responses.

1. (Minor) Page 3, 1st paragraph : *There should be some consistency in the notation of the oxygen isotope. It is written as ^{16}O and O^{16} .*

Response: We thank the reviewer for pointing this out. In the revised manuscript, we have corrected the notation of " O^{16} ", and changed all the related notations into a consistent format " ^{16}O ".

2. Page 5, 1st paragraph: *Regarding the claims of "In the supplementary information we confirm that such fading of replica band intensity from sample #6 to #1 is neither due to sample quality (Fig. S5) nor due to the smear out of the spectral weight (Fig. S6)" and "The origin may be related to slight variations in interfacial bonding conditions, such as differences in bond disorder between FeSe and STO (ref. 30)", the authors analyzed the background intensity in Fig. S5 and showed that there is no correlation between the background intensity and SC gap size, in an attempt to show that the relatively low quality ARPES data in Fig. 2 is not from simple sample quality problem. However, bond disorder may be regarded as part of sample quality. In fact, the data in Fig 2 seems to be dependent on such disorder. Therefore, the claim that the data quality variation is not from the sample quality may not be technically right. Authors may need to reword their claim as it may confuse the readers.*

Response: We agree with the reviewer that the word “quality” itself is qualitative and not precise, which may confuse the readers.

The quality of FeSe can be quantitatively represented by the single-particle scattering rate Γ_1 from superconducting gap fits (Supplementary text “The Determination of the superconducting gap and single-particle scattering rate” and Supplementary Fig. S9). Impurity scattering can drastically reduce the lifetime of the quasiparticles, resulting in broadening of the ARPES lineshapes and diminishing of the quasiparticle peaks, i.e., enlarging Γ_1 . All the comparative data are from samples with nearly identical Γ_1 's (Fig. S9). However, in these samples we observe large variations both of the replica band intensity ratio (Fig. 2) and the superconducting gap of FeSe (Fig. 3). Since these features show no correlation with Γ_1 (Fig. S5(E) and Fig. S9), their variations are likely induced by other issues rather than the FeSe film itself. In such an interfacial system, the atomic registry between FeSe and STO and the bonding distance are important degrees of freedom, which could vary in different films (Ref. 30). [Redacted], TEM studies from our collaborators actually show variation of bonding distance related to the interfacial atomic registry in Regions I and II of the sample ([Redacted]). The electron-phonon coupling strength measured by ARPES is averaged over a square-millimeter-sized light spot, and the relative populations of these different types of interfaces may vary among the samples, leading to the variation of the interfacial electron-phonon coupling strength and the enhancement of superconductivity.

[Redacted]

[Redacted]

We agree with the reviewer that such interfacial structural differences can be considered another aspect of “sample quality” in a broader sense. Therefore, in the revised manuscript, we have reworded all instances of “sample quality” to “single-particle scattering rate in the FeSe layer” to make the claim more rigorous and avoid confusion.

3. Page 8, 1st paragraph: “if there is an oxygen isotope effect on the gap as in the usual BCS formula, it is presumably less than ~ 0.2 meV.”, In obtaining the value of 0.2 meV, they used $\Delta_{interface}$. $\Delta_{interface}$, however, is not clearly defined. I assume it is the gap difference between K-doped and #6 sample but it should be defined. In addition, how do authors know if the expression used in obtaining 0.2 meV is reasonable? Any reference for it

would be useful.

Response: We thank the reviewer for pointing this out. $\Delta_{\text{interface}}$ is the superconducting gap enhancement from the interfacial electron-phonon interaction. Considering the variation of the interfacial electron phonon coupling strength among samples, the value of $\Delta_{\text{interface}}$ ranges from 0.8 to 2.8 meV for Δ_1 , and from 0.75 to 3.85 meV for Δ_2 in our samples. The expression $\left(\sqrt{m_{18\text{O}}/m_{16\text{O}}} - 1\right) \times \Delta_{\text{interface}} \sim 0.2\text{meV}$ was based on a presumption that the interfacial electron-phonon interaction enhances the superconducting gap in a BCS manner, i.e., the isotope coefficient for the gap enhancement from the interface oxygen phonon is $\frac{1}{2}$. In this case, the difference between the gap enhancement in the ^{16}O and ^{18}O samples would have an upper limit $\left(\sqrt{18/16} - 1\right) \times 3.85\text{ meV} \sim 0.2\text{ meV}$, which is beyond our energy resolution.

We agree with the reviewer that the above expression is not a strict form. To make it more rigorous as suggested by the reviewer, in the revised manuscript, we have revised this part and estimate the gap variation based on the literature.

1. According to New J. Phys. 18 (2016) 022001, if the superconductivity of FeSe/STO is due to multi-channel pairing, i.e., the interfacial electron-phonon interaction boosts the superconductivity in FeSe/STO, the isotope coefficient should be less than $\frac{1}{2}$. In this case, the oxygen isotope dependence on the gap should have an upper limit of $\left(\sqrt{m_{18\text{O}}/m_{16\text{O}}} - 1\right) \times \Delta$. Considering a gap variation of 10meV to 13.3meV in ^{16}O samples, the isotope dependence of the superconducting gap should be less than 0.6meV \sim 0.8meV. This is on the edge of the resolution limit of our experimental setup ($\pm 0.35\text{meV}$, i.e. 0.7meV in total).

2. A recent paper suggests that *in the forward scattering mechanism, the isotope effect on T_c is actually to leading order absent* (L. Kulic and O. V. Dolgov, New J. Phys. 19 013020(2017))

In either case, the isotope effect of the superconducting gap is not observable given the current experimental resolution.

In the revised manuscript, we have added the references and the discussion “The data points of the sample #isotope_18 roughly follow the relation of gap versus interfacial electron phonon coupling strength of all the ^{16}O samples, indicating no observable isotope effect on the superconducting gap (Fig. 4). This can be explained by an isotope coefficient $\alpha < 0.5$ due to multi-channel pairing (cite New J. Phys. 18 (2016) 022001). Moreover, a recent theory based on a forward scattering EPI mechanism predicts the superconductivity should be mass-independent to leading order. (cite L Kulic and O V Dolgov, New J. Phys. 19 013020(2017)). Therefore, in these frameworks the gap variation with isotope could be too small to be resolved”.

Reviewer #2 (Remarks to the Author):

This manuscript seeks to address the question of where electron phonon coupling at the interface between FeSe films and the chosen substrate serves to enhance the development of higher Tc superconductivity in the chalcogenide film. By using isotopic substitution the authors provide compelling evidence that that may well be the case in what appears to be a careful study. My feeling is that the manuscript can be published but I am left with some questions.

Response: We thank for the reviewer for his/her recommendation.

1. I was intrigued by the observation that the intensity of the phonon sidebands did not appear to have the Poisson distribution associated with Franck-Condon like processes.

Response: We thank the reviewer for this comment. Since $\lambda < 1$ in FeSe, the intensity drops significantly with increasing n according to the Poisson distribution associated with Franck-Condon like processes $\frac{I_n}{I_0} \sim \frac{\lambda^n}{n!}$. Due to the background and current signal-noise ratio, we can only see the first order replica band, while the higher order Frank-Condon shake-off states are buried in the noise. This is somewhat different from the classic spectra of gaseous hydrogen, in which the electron-phonon coupling is very strong and fully-formed Frank-Condon shake-off states can be observed. Since we can only determine I_1 and I_0 in the current experimental setup, we cannot decisively determine whether the intensity of the phonon sidebands obeys or deviates from the Poisson distribution.

2. The preparation of the samples with different isotopes involved considerable annealing, even before the final ARPES measurement. Are the authors sure that they still had the same isotopically substituted substrate?

Response: We agree with the referee that it is a critical concern whether the isotope substitution remains after the annealing processes. Nevertheless, we are sure that the STO is still isotopically substituted, at least at the interfacial layers. The reasons are list below:

1. We have measured the secondary ion mass spectroscopy (SIMS) on a typical sample after all the high temperature annealing processes (up to $\sim 780^\circ\text{C}$) and just before the growth of FeSe. The results are shown in Supplementary Fig. S1, suggesting that both the surface layers and the bulk of Nb: STO are ^{18}O isotope substituted and the top layers of STO contain an ^{18}O concentration higher than 47%. The FeSe growth and annealing involve much lower temperature heating ($T < 548^\circ\text{C}$) in high vacuum ($P < 5\text{E-}9\text{mbar}$). The oxygen partial pressure is very low, and there is no possibility that the residual ^{16}O in the vacuum leads to any effective substitution of the ^{18}O in STO during the annealing process.
2. Even if the annealing process at around 540°C causes any diffusion of oxygen in bulk STO and slightly reduces the concentration of ^{18}O in the top layer, there is still a

considerable concentration of ^{18}O at the interface. EELS studies were performed after several rounds of long-time annealing before ARPES measurements and an additional annealing at 450°C for 6 hours to remove the Se capping layer in the EELS preparation chamber, and the results clearly show different phonon energies between ^{16}O and ^{18}O samples. Moreover, the statistical analysis of the ARPES data on 18 samples demonstrates different replica band separation energies between ^{16}O and ^{18}O samples. E_S and E_S^* are approximately proportional to the inverse square root of the oxygen masses (Fig. 1F). The isotope dependence of Ω and $E_S(E_S^*)$ can only be explained if the interface still contains a considerable concentration of isotope substitution.

We have added the above discussion into the revised supplementary Information.

3. The superconducting gaps associated with O16 and O18 samples differed by maybe 2.0 meV. The energy resolution in the experiment is 7.5 meV according to the authors. It is not impossible to resolve such a small difference with that resolution but it is on the edge.

Response: We are aware of the small energy difference. This is why it was necessary to repeat the experiment so many times to build up appropriate statistics on nearly-identical samples, and to take extreme care to control and exclude all other possible parameters. Nevertheless, as shown in Fig. 4, as the data points of the sample #isotope_18 roughly follow the relation of gap versus interfacial electron phonon coupling strength of all the ^{16}O samples, indicating no observable isotope effect on the superconducting gap in the current setup.

The English could be improved in places. For instance I was not sure how to interpret the statement "Since the intensity ratio between g' and g reflects the interfacial EPI constant between the FeSe electrons and FK1 phonon, 21-25, our results suggest the variation of interfacial EPI among sample #1-#6." in the first paragraph of page 5.

Further the figure captions are not always clear in their message. They need some work to make them easier to read.

Response: We thank the reviewer for pointing these out. In our revised manuscript, we have changed "our results" to "the variation of the replica band intensity ratio" to avoid confusion. We have also improved the English and tried to make the captions more clear with the help of a native English speaker.

We thank the reviewer for his/her valuable comments, which helped us to make our discussion more comprehensive.

Reviewer #3 (Remarks to the Author):

The authors have performed systematic measurements of the variation in electron-phonon coupling strength in several FeSe/STO samples. They show unambiguously that e-ph coupling is related to the observed 'replica' bands seen in ARPES, and that the e-ph coupling linearly enhances the superconducting gap. This provides strong evidence that besides the doping and the intrinsic bulk FeSe superconducting mechanism, electron-phonon coupling is relevant in monolayer FeSe on STO. I think the paper should be published quickly as it is extremely timely and relevant to the field, and will interest the whole superconductivity community.

Response: We appreciate the reviewer's strong recommendation.

I think there is only major issue that needs to be addressed more carefully. The authors have 6 samples with an oxygen-16 substrate, with "slightly different growth/annealing conditions", leading to a "variation of interfacial EPI". What is the origin of this? How can a few degree change in annealing temperature lead to a dramatic change EPI (of more than a factor 2)? Can the authors elaborate on this point?

Response:

We thank the reviewer for the constructive comments. We agree with the reviewer that the "slightly different growth/annealing conditions" should not be the direct origin of the variation of the interfacial EPI. While preparing the samples, even if we keep the growth/annealing conditions identical (comparing sample #1 and sample #5 for instance), the resulting samples show dramatically different EPI strength. The difference may come from subtle details at the substrate surface, for example, the step width, stoichiometry, and complicated surface reconstructions of STO, which could affect the single-layer FeSe grown on it despite almost identical growth/annealing conditions. Our data show that although the FeSe films have the same carrier density, chemical composition, and single particle impurity scattering rate, the interfacial electron-phonon coupling strength can change. Minor differences at the interface itself can have large effects on the coupling of phonon modes across the interface. For example, in such an interfacial system, the atomic registry between FeSe and STO and the bonding distance are important degrees of freedom, which could vary among the films (ref.30). [Redacted], TEM studies by our collaborators actually show variations in bonding distance related with interfacial atomic registry in Regions I and II of the sample ([Redacted]). The electron-phonon coupling strength measured by ARPES averages over a square-millimeter-sized light spot, and the relative populations of these different registries of interfaces may vary among the samples, leading to the variation of the interfacial electron-phonon coupling strength. A rigorous analysis will require in-situ ARPES and in-situ interfacial structural characterization on the same region of a single sample.

[Redacted]

[Redacted]

At present, we can achieve different interfacial EPI by growing a large number of samples and analyzing their replica bands. However, the interfacial EPI on each individual sample is not known before measuring ARPES, even if we precisely control the growth/annealing condition. In our revised manuscript, we have reworded the expression “prepared with slightly different growth/annealing conditions” to “prepared with similar growth/annealing conditions” to avoid confusion. Besides, we have added “The origin may be related to slight variations of STO surface which leads to different interfacial bonding conditions, such as differences in bonding registry between FeSe and STO (ref. 30) “to make it more clear.

Some minor issues:

1. *In the discussion the authors mention that the oxygen isotope effect on T_c should be unobservable. I think it is worth mentioning that in the forward scattering mechanism, the isotope effect on T_c is actually to leading order absent, see for example Kulic&Dolgov in <http://iopscience.iop.org/article/10.1088/1367-2630/19/1/013020/meta>.*

Response: We thank the reviewer for pointing this out. In the revised manuscript, we have added this reference and a corresponding discussion on the oxygen isotope effect on T_c .

2. *As for isotope substitution: It is exciting to see that they found a change in the replica band location upon substitution of oxygen-16 to -18 in only the top layers of the substrate. I do wonder to what extent the possible ferro-electric transition in STO, that exists upon oxygen substitution in the bulk (Rowley Nature Physics 10, 367 (2014)), is relevant for these samples. Can the authors elaborate on this relation?*

Response:

We thank the reviewer for raising this point, as the ferroelectric property of STO substrates is indeed a critical issue for understanding the behavior of STO-related interfaces. However, the STO was heated to high temperature ($\sim 780^\circ\text{C}$) before FeSe growth and also doped with Nb, so that it is conducting. The itinerant electrons would screen the electric field of STO if there is any ferroelectricity. We do not think possible variation of the dielectric constant upon isotope substitution is related to the observed replica band energy shift or superconducting gap based on the following reasons:

1. Previous mutual inductance and Raman studies suggest that the FeSe/STO interface shows a softened ferroelectric phonon mode below 50K (Y-T Cui, et.al, PRL 114, 037002, 2015). According to this scenario, displacements of positive and negative charge centers happens near the interface, even without ^{18}O substitution. There is no turning on/off of the displacement at the interface in ^{18}O samples compared with ^{16}O samples.
2. The observed replica band energy shift is approximately proportional to the inverse square

root of the oxygen mass. This can be explained by the isotope dependence of the phonon modes. No additional effect needs to be included.

3. In Fig.4, the data point of the ^{18}O sample roughly follows the relation of gap versus interfacial electron phonon coupling strength of the ^{16}O samples, indicating no observable isotope effect on the superconducting gap. This is consistent with the absence of an isotope effect on T_c based on a forward scattering EPI mechanism (L Kulic and O V Dolgov, New J. Phys. **19** 013020(2017)) or an isotope coefficient less than $\frac{1}{2}$. No additional effect needs to be included.

In the revised manuscript, following the reviewer's suggestion, we have added "Note that the STO here is conducting due to vacuum annealing and Nb doping, if there were any ferroelectricity in ^{18}O substituted STO as observed in bulk and insulating STO (ref. 36), the electric field would be screened by the itinerant electrons and would not be likely to affect the FeSe/STO interface." Besides, in our Methods part, we have added "The commercial 0.5%wt Nb-doped SrTi16O3 substrates are from Hefei Kejin Materials Technology Co., Ltd. To avoid charging effects, the SrTi $^{16}\text{O}_3$ /SrTi $^{18}\text{O}_3$ grown on commercial substrates is doped with ~0.7%wt Nb" to claim the Nb doping in the samples.

We thank the reviewer for the valuable comments, which helped to make the paper more robust and thorough.

REVIEWERS' COMMENTS:

Reviewer #1 (Remarks to the Author):

I had two major comments on the manuscript. Both of them are comprehensively answered and I am happy with the answers.

1) The first one was on the 'sample quality' issue. The 'sample quality' cannot be clearly defined and now the authors use 'single-particle scattering rate in the FeSe layer' Γ_1 to quantify the quality of the samples. Yet, they find no correlation between Γ_1 and replica band intensity ratio as well as the superconducting gap, supporting the claim.

2) The discussion on the effect of the isotopes on the gap size is revised with appropriate references. As this is an important issue in their argument, it is also important that the issue is cleared up, by clearly showing how they came up with the conclusion and provide appropriate references.

There are some editorial issues (mostly grammatical), these should be taken care of in the later stage if the manuscript is accepted.

Reviewer #3 (Remarks to the Author):

In my opinion the response from the authors to all three referees questions is satisfactory and I can recommend publication.

REVIEWERS' Comments

Reviewer #1 (Remarks to the Author):

I had two major comments on the manuscript. Both of them are comprehensively answered and I am happy with the answers.

1) The first one was on the 'sample quality' issue. The 'sample quality' cannot be clearly defined and now the authors use 'single-particle scattering rate in the FeSe layer' Γ_1 to quantify the quality of the samples. Yet, they find no correlation between Γ_1 and replica band intensity ratio as well as the superconducting gap, supporting the claim.

2) The discussion on the effect of the isotopes on the gap size is revised with appropriate references. As this is an important issue in their argument, it is also important that the issue is cleared up, by clearly showing how they came up with the conclusion and provide appropriate references.

Response: We appreciate the reviewer's positive comments.

There are some editorial issues (mostly grammatical), these should be taken care of in the later stage if the manuscript is accepted.

Response: We thank the reviewer for his/her comments on editorial issues, which helped us to improve our manuscript.

Reviewer #3 (Remarks to the Author):

In my opinion the response from the authors to all three referees questions is satisfactory and I can recommend publication.

Response: We appreciate the reviewer's positive remarks and recommendation.